# Finding Species-Specific Extracellular Surface-Facing Proteomes in Toxic Dinoflagellates

**DOI:** 10.3390/toxins13090624

**Published:** 2021-09-06

**Authors:** Kenrick Kai-yuen Chan, Hang-kin Kong, Sirius Pui-kam Tse, Zoe Chan, Pak-yeung Lo, Kevin W. H. Kwok, Samuel Chun-lap Lo

**Affiliations:** 1Department of Applied Biology and Chemical Technology, Faculty of Applied Science and Textiles, The Hong Kong Polytechnic University, Hung Hom, Hong Kong; kai-yuen.ky.chan@connect.polyu.hk (K.K.-y.C.); hang-kin.kong@polyu.edu.hk (H.-k.K.); pktse@polyu.edu.hk (S.P.-k.T.); zoe-z.c.chan@polyu.edu.hk (Z.C.); pypatlo@polyu.edu.hk (P.-y.L.); kwh.kwok@polyu.edu.hk (K.W.H.K.); 2Research Institute for Future Food, The Hong Kong Polytechnic University, Hung Hom, Hong Kong

**Keywords:** proteomics, toxic dinoflagellates, cell surface labeling, orthogroup inference

## Abstract

As a sequel to our previous report of the existence of species-specific protein/peptide expression profiles (PEPs) acquired by mass spectrometry in some dinoflagellates, we established, with the help of a plasma-membrane-impermeable labeling agent, a surface amphiesmal protein extraction method (SAPE) to label and capture species-specific surface proteins (SSSPs) as well as saxitoxins-producing-species-specific surface proteins (Stx-SSPs) that face the extracellular space (i.e., SSSPs_Ef_ and Stx-SSPs_Ef_). Five selected toxic dinoflagellates, *Alexandrium minutum*, *A. lusitanicum, A. tamarense, Gymnodinium catenatum,* and *Karenia mikimotoi,* were used in this study. Transcriptomic databases of these five species were also constructed. With the aid of liquid chromatography linked-tandem mass spectrometry (LC-MS/MS) and the transcriptomic databases of these species, extracellularly facing membrane proteomes of the five different species were identified. Within these proteomes, 16 extracellular-facing and functionally significant transport proteins were found. Furthermore, 10 SSSPs and 6 Stx-SSPs were identified as amphiesmal proteins but not facing outward to the extracellular environment. We also found SSSPs_Ef_ and Stx-SSPs_Ef_ in the proteomes. The potential functional correlation of these proteins towards the production of saxitoxins in dinoflagellates and the degree of species specificity were discussed accordingly.

## 1. Introduction

There are concerns about the possible contamination of seafood with paralytic shellfish toxins (PSTs). Some of these PSTs are produced by toxic dinoflagellates, common causative agents of harmful algal bloom (HAB). PSTs would accumulate along food chains, causing paralytic shellfish poisoning (PSP) and could lead to large-scale fish death and massive economic loss. Among the different PSTs, saxitoxins (STX) are the most common. They are potent sodium channel blockers of neurons with LD50 as low as 5.7 μg/kg for humans [1]. The bioaccumulation of STX can be seen in filter-feeding bivalves, moon snails, lobsters, and crayfish, etc. [2]. STX would eventually be passed onto humans and may cause deadly food poisoning cases. Notably, the distribution as well as the frequency of occurrence of PSP cases keep broadening and increasing globally, respectively. More than 2000 cases of intoxication are reported per year and 15% of these resulted in death [3]. Owing of the possible health risk and massive economic damages due to the occurrence of large scale HAB, it is important to rapidly identify its causative agent(s). However, current morphological methodologies cannot distinguish morphologically similar toxic dinoflagellates from non-toxic ones. Consequently, in most HAB studies that involve the biochemistry of dinoflagellates, Internal transcribed spacer (ITS)-regions between 18S to 5.8S and 5.8S to 28S ribosomal RNA would be sequenced to confirm the identities of various dinoflagellates used. However, there is still confusion about the true identities of some dinoflagellate members within the *Alexandrium* superfamily because some of the earlier DNA sequencing results might have been obtained using wrongly identified (via morphological methodologies) reference cultures. From another perspective, Murray S. et al. were the first to document the use of quantitative PCR methodologies to identify SxtA4 gene as an indicator of toxicity [4]. Of course, this is not the same as the identification of the STX-producing dinoflagellates and there are dinoflagellates that carry the SxtA4 gene but do not produce STX (e.g., Tasmanian ribotype of *Alexandrium tamarense* [4]).

In order to identify toxic dinoflagellates rapidly and objectively, we developed an objective protein/peptide expression profile (PEP)-based rapid species identification method for HAB-causative dinoflagellates using matrix-assisted laser desorption/ionization time-of-flight mass spectrometry (MALDI-TOF-MS) [5,6]. It was modified from the PEP approach for the identification of bacteria [7,8,9]. Unique PEP fingerprints were found among different dinoflagellates and cyanobacteria [5,6,10]. This implies that different toxic dinoflagellates may process different species-specific-proteins (SSPs) that are discoverable by the MALDI-TOF-MS methodology. However, to further study fragments of SSPs in the mass spectrum, one has to perform de novo sequencing on these fragments to obtain their amino acid sequences in their backbones. A relatively large amount of SSP fragments with high intensity peaks have to be loaded onto the target plate, and the mass spectrometer has to have exceptionally high sensitivity. It is currently difficult to perform de novo sequencing on all individual peaks of interest on the PEP fingerprints to obtain their amino acid backbone sequences for identification purposes. Furthermore, it should also be stressed that MALDI-TOF-MS is a sophisticated instrument with stringent sample preparation procedures, and its application for on-site species identification is difficult. Thus, the identification of species-specific surface proteins (SSSPs) for the development of a simple antibody-based identification method is probably the most practical approach. In addition, we defined that SSSPs referred to the proteins that existed on the external surface with relative species specificity. For those SSSPs with part of their structures exposed to the external environment (i.e., readily accessible from the external environment), they should be called SSSPs_Ef_. To this end in order to achieve the aim of identifying SSSPs and SSSPs_EF_, documentation of the proteomic profiles of amphiesma from different species could give distinctive cell-surface biomarkers that could possibly be used for species identification purposes. However, two enabling platforms are needed for such an operation. Firstly, to enable successful and accurate identification of SSSPs, bioinformatic support using a comprehensive genome or transcriptome database is needed. However, due to the massive genome sizes of most toxic dinoflagellates, which are predicted to range from 3 to 245 Bbp [11], it is currently technically very difficult to construct a genome database for regular dinoflagellates. Secondly, we need a method that allows us to label and capture SSSPs_Ef_. To provide the first enabling platform, we selected five toxic dinoflagellate species and built their transcriptomes instead. The five species were four STX-producing species: *Alexandrium minutum* (CCMP113), *A. lusitanicum* (CCMP1888), *A. tamarense* (ATCI03), and *Gymnodinium catenatum* (CCMP1937) and an STX-negative species, *Karenia mikimotoi* (K1). For the second platform to look for SSSPs_Ef_, we looked at the amphiesma that is exclusively found in dinoflagellates. In addition, the amphiesma is a cellular boundary ultrastructure composed of an outermost membrane called the pellicular layer and a cytoplasmic membrane [12,13]. Some species of dinoflagellates possess a thecal structure between the pellicular layer, and the outermost membrane and these are called “thecate” dinoflagellates [13,14]. Apart from identifying SSSPs_EF_, finding extracellular surface-facing proteomes may also enhance our understanding of the transporters that are localized on the cell surface of these dinoflagellates. Previous phycological studies have suggested that the amphiesma could play an essential role in a wide range of cellular functions, such as cell signaling, the entry of nutrients and the transportation of various molecules in and out of dinoflagellates [12,15]. Especially in STX-producing dinoflagellates, whether the amphiesma takes part in the export of STX is worth studying. Although previous molecular studies have revealed the partial similarity of STX biosynthetic mechanism between cyanobacteria and dinoflagellates [16,17,18], there is insufficient proteomic evidence to confirm that the protein products of these putative genes, especially the putative STX transporter (i.e., gene product of SxtF/M), exist in dinoflagellates for the extrusion of STX. Additionally, the exact subcellular location of this putative STX transporter could be an important concern. To address these questions, the investigation of cell surface proteomes and hence the extracellular-facing transporters of toxic dinoflagellates should be essential.

There were several earlier trials that were conducted to extract cell wall proteins in dinoflagellates by cold shock [19], and membrane proteins from the insoluble cellular fraction of the homogenate [20,21]. Both methods took advantage of the low solubility of amphiesmal proteins in an aqueous environment and used methods modified from the cyclohexane-diamine tetraacetic acid (CDTA)-based plant cell wall protein extraction protocol [22,23]. Both methods captured proteins indiscriminately without considering whether they were on the external or internal sides of the amphiesmal membrane. Furthermore, because of the lack of appropriate genome databases for bioinformatic searches at that time, the accuracy of their finding as well as correct identification of SSSPs_Ef_ might have been compromised. To this end, we employed a cell membrane impermeable, cleavable, water-soluble, and stable isotopic labeling reagent called sulfosuccinimidyl-2-(biotinamido) ethyl-1,3-dithiopropionate (sulfo-NHS-SS-biotin) to capture external-facing amphiesmal proteins. As illustrated in Figure 1, this reagent was expected to selectively label SSSPs_Ef_ as the chemical could not get past the plasma membrane readily.

In the labeling chemical, the N-Hydroxysuccinimide (NHS) part of the labeling reagent provided reactivity towards the primary amine and the negatively charged sulfo- group prevented the tag from passing through the phospholipid plasma membrane bilayer. Therefore, we expected that proteins on the cell surface could be labeled following the designed reaction scheme (Appendix A) and subsequently those labeled SSSPs_Ef_ and Stx-SSPs_Ef_ could then be identified. Whole cell lysate (WL) and insoluble cellular fraction (ICF) were prepared to act as the control in the study.

## 2. Results

### 2.1. Molecular Identification and Quantification of STXs for the Selected Dinoflagellate Species

To confirm the identities of the toxic dinoflagellates used, their ITS regions between 18S to 5.8S and 5.8S to 28S ribosomal RNA (rRNA) were amplified and sequenced using the PCR method. As shown in Table 1 below, the BLAST search results confirmed that the identities of CCMP113, CCMP1888, ATCI03, CCMP1937 and K1 were *Alexandrium minutum, A. lusitanicum, A. tamarense, Gymnodinium catenatum* and *Karenia mikimotoi,* respectively.

In addition, the presence of STXs in the five species was validated by LC-MS/MS approach. By referring to the standard of 12 STX-derivatives (i.e., STX, dcSTX, neoSTX, GTX1-5, C1-2 and dcGTX2-3), the intracellular concentration of STXs in the five species was quantified. As shown in Figure 2, CCMP1937 showed the highest intracellular concentration of STXs (75.4 fg/µm^3^). Among the Alexandrium species, CCMP113 and CCMP1888 had similar intracellular concentration of STXs, which were 2.77 fg/µm^3^ and 2.35 fg/µm^3^, respectively. Their toxicities were relatively low compared to ATCI03 (9.95 fg/µm^3^). None of the 12 STX-derivatives were found in K1. Therefore, CCMP113, CCMP1888, ATCI03 and CCMP1937 were confirmed to be STX-producing species while K1 was not an STX-producing species.

### 2.2. Transcriptomic Libraries and Orthogroup Inference

As no public genome database is available for the bioinformatic search for the five selected dinoflagellate species, in-house cDNA libraries of the five species (NCBI Bioproject Accession no.: PRJNA634431) were built to support the proteomic studies and protein identity search. Peptide ions were searched with reference to these in-house cDNA databases. In addition, gene sequences from these libraries were translated and inputted for the orthogroup inference and the results were used for searching purposes.

The results of the transcriptomic orthogroup inference are presented in Table 2. There were a total of 695,887 translated genes inputted from the transcriptomes of the five selected species, and 55% of them were successfully assigned into orthogroups. There were 86,675 orthogroups, and they could be further classified into species-specific orthogroups (1.78%), STX-producing-species-specific orthogroups (3.70%), single-copy orthogroups (3.73%), *Alexandrium* species-specific orthogroups (12.8%), and orthogroups with all species present (16.5%). In addition. proteins identified as species-specific orthogroups and STX-producing-species-specific orthogroups were regarded as SSPs of interest.

As shown in Table 3, pairwise orthogroup comparison and interspecies phylogenetic distance were interrelated. CCMP113 and 1888 shared 95% of overlapped orthogroups, which indicated relatively high similarity between the transcriptomes of the two species. Conversely, ATCI03 and CCMP1937 were the most dissimilar pair, with only 53% of common orthogroups.

### 2.3. Efficiency of Our Novel Method in Labeling Amphiesmal Protein

As previously mentioned, we employed a sulfo-NHS-SS-biotin and novel surface amphiesmal protein extraction (SAPE) method to selectively label and capture SSSPs_Ef_. A set of controls, extracts of whole cell lysate (WL) and insoluble cellular fraction (ICF) were also obtained to compare with our SAPE methodology in terms of diversity and efficiency in amphiesmal protein enrichment. We also tried the cold shock approach [19] to enrich the cell wall proteins in dinoflagellate species, but since unwanted extensive cell rupture in fragile dinoflagellates such as K1 was induced, it was not studied any further. Eventually, 15 sets of proteomes (3 replicates × 5 species) were generated.

Compared with the ICF and WL methods, we would like to highlight that samples of SAPE showed not only an improvement in the percentage yield of amphiesmal protein (Table 4), but also gave higher consistency in terms of the principal components (Appendix A). There were proteins in the SAPE samples of each species successfully labeled with a 145.020 Da mass add-on. The mass add-on was specifically located at the exposed lysine residues of these proteins (Appendix A). These tagged proteins were expected to face the external region away from the plasma membrane. Together with the GO annotation, subcellular locations of proteins identified in SAPEs could be further confirmed. It was observed that a total of 1015 proteins in the SAPEs of the five species were annotated with amphiesma-related GO terms and 167 of them were labeled. However, we noted that the labels were also found in some non-amphiesmal proteins such as ribosomal proteins and elongation factors. This indicated that a small portion of the labeling reagent could permeate the plasma membrane to some extent which resulted in this non-specificity.

Sixteen labeled amphiesmal proteins were found to belong to homologs of transporter proteins (Table 5).

As listed in Table 5, the majority of the homologs of transporter proteins identified with our labeling strategy were related to energy production (i.e., ATP synthase and NAD(P) transhydrogenase) or protein transport (i.e., clathrin and protein translocase). There were three transporter homologs that were commonly identified in multiple species (Table 6). Both ATP synthase (also known as F-type ATPase) and NAD(P) transhydrogenase were ATP-dependent and proton pump transporters. Apart from that, clathrin was functionally annotated as an endosomal protein that might take part in the transport of transferrin, receptor internalization and membrane organization. In addition, the homolog of protein translocase SecA was solely found in the SAPEs of ATCI03. Protein translocase SecA was an ATP-dependent transporter responsible for the import of polypeptides and proteins across the membrane. In addition, no homologs of the putative STX transporter were labelled and identified in the SAPEs of STX-producing dinoflagellates studied.

To a certain extent, the findings validated the efficiency and capability of our amphiesmal protein labeling method. Although these homologs of transporters were functionally recognizable and confirmed to be localized on the cell surface, they might not be specific enough or exist in sufficient quantities for species discrimination. Therefore, proteins identified in SAPEs were further analyzed by orthogroup clustering (see below).

### 2.4. Proteomic Orthogroup Analysis for the Surface Amphiesmal Protein Extracts

Proteome-scaled orthogroup clustering was carried out for cross-species comparison. Due to the specialty of SAPE, the results of orthogroup assignment for the proteomes of SAPE would serve as the basis for discovering SSSP and SSSP_Ef_ orthologs that were expressed by certain species. Protein sequences that had surpassed the default threshold of reciprocal best normalized hit were assigned to different orthogroups.

As shown in Table 7, K1 contained the largest number of proteins that could not be assigned to any orthogroups. Species-specific orthogroups indicated the possibility of one species containing a distinctive protein biomarker that could be species-discriminating, whereas the orthogroups shared by the four STX-producing species could reveal some common biological functions and biomarkers among these species. Therefore, species-specific orthogroups and STXs-producing-species-specific orthogroups were the targets of interest of this study. We reported that 10 SSSPs (Section 2.4.1) and 6 Stx-SSPs (Section 2.4.2) were revealed.

#### 2.4.1. Species-Specific Protein Orthologs

As shown in Table 8, we noted that the dual specificity mitogen-activated protein kinase-like ortholog identified in SAPE of K1 was successfully labeled on the peptide “GKAQMGTYADNLGAGSHSGGGVTEAPR”(Appendix A). It could be a possible SSSPs_Ef_ and could be used as a biomarker for distinguishing K1 from the other four STXs-producing species studied. Still, the majority of the K1-specific orthologs were homologous to chromophore-linking proteins including subunits of fucoxanthin-chlorophyll a-c binding protein and chlorophyll a-b binding protein. Compared to the four STX-producing species, this observation implied the possibility of K1 containing different plastid and chromophore compositions in its light harvesting system. These proteins were annotated as transmembrane in nature. However, they were unlabeled. Thus, we predicted that they could be embedded in inaccessible regions on the membrane and hence were not labeled. Apart from K1, a homolog of isocitrate dehydrogenase (NADP) (P16100) was identified as an SSP of CCMP113. Isocitrate dehydrogenase was known to be responsible for the catalytic formation of 2-oxoglutarate in the tricarboxylic acid cycle in mitochondria. Although the homolog was identified as an SSP for CCMP113, the overall protein coverage in MS was so low that only one peptide was recovered. Therefore, further validation would be needed for this homolog. Other than that, a homolog of pyruvate dehydrogenase kinase was revealed in the SAPEs in CCMP1937 as an SSP. Pyruvate dehydrogenase kinase was known to take part in the regulation of glucose metabolism. Yet, the SSPs of CCMP113 and CCMP1937 were neither labeled nor annotated as cell-surface membrane proteins. Therefore, these might exist as cytoplasmic SSPs. We think that the above orthologs could be used as biomarkers for species discrimination among other species studied in this study. Lastly, no SSSPs were identified from the SAPEs of CCMP1888 and ATCI03.

# Estimated by the best score and E-value of the matches in BLASTp results. ^ These peptides contained at least 15 amino acids and were identified with peptide score > 50. Please refer to Appendix A for the peptide score of these peptides. * The amino acid sequences, individual mascot score of each protein identified and the position of labeled residue are shown in Appendix A. Note—Mascot score of all these protein matches reached the confidence level > 95% (i.e., Mascot score > 90).

#### 2.4.2. Stxs-Producing-Species-Specific Orthologs

Apart from the SSSPs, information on the STX-producing-specie-specific protein (Sxt-SSP) and extracellular facing Stx-SSP (Stx-SSP_Ef_) orthologs was of interest and could contribute to our understanding of the common expressions among the four STXs-positive species in this study. Additionally, we studied the difference between the four STX-positive species and non-STX-producing K1.

As shown in Table 9, six Stx-SSPs were successfully enriched by the SAPE method. Functionally, these orthologs could be further grouped into five categories, namely tricarboxylic acid transmembrane transport, nitrogen compound metabolism, light-harvesting, ATP synthesis and protein translation. Interestingly, components of chromophore-linking proteins were also revealed in the orthogroups shared by STX-positive species. This implied that plastids and chromophore compositions in the STX-positive species were similar. This observation also implied the possibility of correlating the light harvesting system to the ability of STX biosynthesis in dinoflagellates. More importantly, the orthogroup containing homologs of chloroplastic F-type H^+^-transporting ATPase subunit alpha was found to be successfully labeled in both STX-producing species. Thus, it was suggested to be a possible Stx-SSP_Ef_ for distinguishing the four STX-positive species from the STX-negative K1. In addition, orthologs of glutamine synthetase in the STX-producing species were identified with low protein coverage. Further investigation to verify our prediction would be required.

# Estimated by the best score and E-value of the matches in BLASTp results. ^ These peptides contained at least 15 amino acids and were identified with peptide score > 50. Please refer to Appendix A for the peptide score of these peptides. * Successfully labeled in all four STX-positive species. The amino acid sequences, individual mascot score of each protein identified and the position of labeled residue are shown in Appendix A. Note—Mascot score of all these protein matches reached the confidence level > 95% (i.e., Mascot score > 90).

## 3. Discussion

Earlier findings have revealed that the amphiesma is involved in a wide range of cellular processes during the cell cycle process through the rearrangement of its amphiesmal layers [25,26,27,28]. Due to the functional diversity of the amphiesma, the contribution of the amphiesma to the STXs-biosynthesis related mechanisms is possible. Therefore, any unknown connections between STX-production and the amphiesma could be revealed by comparing the surface proteomes of the STX-positive dinoflagellates with those of the STX-negative dinoflagellates. Although GO annotation could provide insights into the subcellular location of proteins, some proteins could be annotated with multiple cellular locations. Thus, the use of cell lysate studies alone would be insufficient to tell which proteins really existed on the membrane. In this study, our label-assisted SAPE approach and proteome-scale clustering of ortholog sequences were able to identify the extracellular-facing homologs of transporters and species-discriminating cell-surface proteins.

### 3.1. Extracellular-Surface Facing Transporters

Twenty homologs of membrane transporters from the SAPEs of the five species were successfully labeled and identified (Table 5). We predicted that these proteins could exert their functions within the amphiesmal membrane structures and contributed to the functional diversity of the amphiesma. The majority of the transport proteins identified were homologs of mitochondrial F-type ATPase (also ATP synthase) that could facilitate the transport of hydrogen ions across the membrane structure during its action in ATP hydrolysis or synthesis. Although this F-type ATPase was commonly known to be localized in the inner membrane of mitochondria in eukaryotic cells, our findings that F-type ATPase was also identified in the cell wall of *Alexandrium catenella* were consistent with the results of a previous proteomic study [19]. This, together with the proof that isotopic labels were found on these proteins, further indicated the possibility that these proton pumps were expressed on the amphiesma and exposed to the extracellular environment. Apart from F-type ATPase, NAD(P) transhydrogenase (NNT) was another major type of proton pump identified in the SAPEs of dinoflagellates. NNT was involved in the transhydrogenation between NADP and NADH that was coupled to ATP hydrolysis. During the process, NNT would act as a proton pump. Therefore, we predicted that these proton pumps might contribute to the regulation of electrochemical potential difference across the amphiesmal membrane, and hence the energizing of nutrient influx [29].

For the proteins that were related to nutrient transport, a homolog of the ammonium transporter and clathrin were found in the SAPEs. Besides, we anticipated that homologs of the vesicle-forming clathrin identified could also be associated with the transport of nutrients across the amphiesma. Earlier studies suggested that nutrient transport in some dinoflagellates could mainly be clathrin-dependent endocytosis [30,31] and predicted that the flagellar canal would be the only active site for clathrin-dependent endocytosis [15]. Since the flagellar canal was embedded in the amphiesma, it was possible that the clathrin vesicle could be exposed to the cell surface along this portal and consequently labeled by our technique. We also found a labeled homolog of protein translocase that could be associated with protein trafficking in the SAPEs. However, there was a wide range of possible targets that could be processed by this transport protein, and thus conclusion of how this homolog contributed to amphiesmal transport could not be drawn at this time. Although our labeling approach confirmed that the abovementioned transporters were localized on the cell surface and that they faced the extracellular side of the amphiesma, they were not SSSPs in terms of species specificity as determined by orthogroup inference.

In addition, we were unable to identify or label any homologs of the STX transporter in the SAPEs (i.e., gene product of SxtF/M). Since SxtF/M was not one of the known core genes (i.e., SxtA1-4, SxtB, SxtD, SxtG, SxtH/T, SxtI, SxtS and SxtU) in the predicted STX biosynthetic mechanism of dinoflagellates [32,33,34,35], it was possible that SxtF/M could be missing or not translated as a protein in some STXs-producing dinoflagellates. Thus, we suggested that the putative transporter for STXs might not exist in the four STXs-producing dinoflagellates under investigation.

### 3.2. On-Surface STX-Producing-Species-Specific Proteins

When comparing protein orthogroups between the STX-positive and STX-negatives species, we noted that homologs of fucoxanthin-chlorophyll proteins (FCPs) and chloroplastic F-type H^+^-transporting ATPase (F-ATPase) in STX-producing species were put into STX-species-specific orthogroups while those in STX-negative species (K1) were grouped in separate orthogroups. FCPs and F-ATPase could be found in peridinin-containing dinoflagellates that played an important role in light harvest [36,37] and energy converting behavior [38,39]. This indicated that the four STX-positive species shared similar FCPs and F-ATPase in their plastids, which were significantly different from those in the plastids of K1. This observation could be further explained by earlier phylogenetic studies that the *Karenia* species contained a different type of plastid while *Gymnodinium* and *Alexandrium* species might have the same type of plastid [40,41,42]. Although multiple studies suggested that STX biosynthesis was correlated to photosynthesis and energy production in dinoflagellates [43,44,45], whether the FCPs and F-ATPase contributed to STX biosynthesis is still unclear, and further investigation is required.

Apart from FCPs and F-ATPase, we would like to highlight the two other Stx-SSP orthologs that were revealed in STXs-positive species but were absent in STX-negative K1 (Table 9). They were glutamine synthetase (GS) and mitochondrial dicarboxylate/tricarboxylate transporter (DTC). GS was an enzyme that catalyzed the synthesis of glutamine from ammonia [46,47,48] and acted as one of the essential means for plants to establish tolerance against ammonia toxicity [49]. Previous investigations showed that the tolerance of ammonia stress in microalgae was also a significant factor affecting algal growth and survival [50,51,52,53,54,55]. In addition, the activation of GS was suggested to be indirectly regulated by the amount of 2-oxoglutarate on the transcriptional level [56]. Collectively these implied that the role of DTC (also known as oxoglutarate-malate antiporter) might correlate with the action of GS as DTC controlled the transport of 2-oxoglutarate across the inner mitochondrial membrane in exchange for malate or other tri/dicarboxylic acids, and hence would jointly influence the nitrogen metabolism. It had been suggested that the production of STX was intertwined with nitrogen metabolism [35,57]. As a key enzyme involved in nitrogen metabolism, we predicted that the expressions of GS and DTC in the STX-producing species were different from those in the non-STX-producing K1. This difference might allow a higher capacity of STX-producing species to manipulate ammonia toxicity from internal metabolic byproducts of nitrogen-containing compounds such as STXs.

In addition, the remaining common orthogroup of pentatricopeptide repeat-containing proteins revealed in STXs-producing species was related to protein translation. Similar to nitrogen metabolism, an earlier study also revealed that the expressions of protein translation-related proteins were specifically upregulated in the STX-producing species during the biosynthetic process of STXs [36]. The occurrence of the translation-related orthogroup of the STXs-producing species might be consistent with this previously discovered correlation of protein translation to the production of STXs. However, the exact function of pentatricopeptide repeat-containing proteins is still unclear, and thus more supporting evidence of this notion is needed.

### 3.3. On-Surface Specific Proteins

An SSSP_Ef_ for K1 and an Stx-SSP_Ef_ for the four STX-producing species were successfully labeled and identified. For the Stx-SSP_Ef_, homologs of chloroplastic F-ATPase were specifically revealed on the cell surface of the four STX-producing species. With reference to the results of orthogroup inference, please note that this chloroplastic F-ATPase did not show any orthologous relation with the mitochondrial F- ATPase discussed. Chloroplastic F-ATPase was a membrane enzyme responsible for ATP production and proton transport that was present in the thylakoid of green algae [58]. However, our results showed that homologs of this protein were consistently labeled in all four STX-positive species, which again, further implied that these four STX-producing species could possess a common ATP synthetic mechanism within their amphiesmal region. Furthermore, this discovery suggested that it could be a usable cell-surface biomarker for distinguishing the four STX-positive species from K1. For the SSSP_Ef_ of K1, a homolog of dual specificity mitogen-activated protein kinase 4 was successfully labeled by the isotopic tag in K1 (Appendix A). This indicated that this protein could potentially be a surface specific biomarker for distinguishing K1 from the other four species as well. Functionally, this protein takes part in various mammalian cellular responses against stresses [59]. Although the SSSPs identified could be ubiquitous in other species that were not part of this study, our findings supported the concept of looking for cell-surface species discriminating biomarkers by comparing the surface proteomes among harmful dinoflagellates.

### 3.4. Advancements Made in Labeling and Extraction Techniques

As highlighted, the advancements made in our cell-surface labeling technique and membrane protein extraction method helped reveal a larger population of amphiesmal proteins from the innermost plasma membrane of the cell surface of dinoflagellates. Compared with the previous cell wall/thecal protein extraction method for dinoflagellates [19,20,21], our SAPE in this study significantly increased the yield of amphiesmal proteins by four-fold. Although its efficiency in identifying labeled peptides was limited by some intrinsic problems such as the difficulties in ionizing peptides that were labeled, state-of-the-art technology of LC-MS has already been applied to minimize these limitations. The scope of cell-surface proteomics was better defined after the application of our labeling method as it eliminated the interference from abundant cytoplasmic proteins [60]. Due to the depletion of abundant proteins from the cytoplasm, a more distinctive proteomic profile could be generated, which was helpful to the development of non-morphological methods and more objective methods for the identification of toxic species. In the future, better documentation with better technologies on species-specific extracellular-facing surface proteomes would support the ultimate discovery of representative biomarkers in toxic dinoflagellates. The findings of this study may just be a starting point that would contribute to the future development of surface protein-based detections, such as the development of antibodies for the quick identification of SSSPs.

## 4. Materials and Methods

### 4.1. Culture Conditions

Five selected species, *Alexandrium minutum* (CCMP113), *A. lusitanicum* (CCMP1888), *A. tamarense* (ATCI03), *Gymnodinium catenatum* (CCMP1937) and *Karenia mikimotoi* (K1), were cultured in L1-Si algae growth medium [61] with 12:12hr light-dark cycle at 21 °C. Sterile synthetic seawater was prepared by 29.2 g/L of Instant Ocean^®^ sea salt (Spectrum Brands, Madison, WI, USA).

Cell density of the cultures on day 0, 3, 6, 9, 12, 15, 18, 21, 24 and 27 was quantified by Sedgewick-Rafter counting chamber and respective growth curve of each species was plotted accordingly. Cultures were cultivated starting with cell density at 4000 cells/mL and harvested at their stationary phase by centrifugation at 8000× *g* for 10 min at room temperature. Supernatants were discarded and the cell pellets were resuspended in specific buffers for different subsequent applications.

### 4.2. Molecular Identification

High Pure PCR Template Preparation Kit (Hoffmann-La Roche, Basel, Switzerland) was used to obtain the genomic DNA from dinoflagellates and standard protocol from the manufacturer was followed. The extracted DNA was eluted in 30 µL of nuclease-free water. After that, thirty-five cycles of polymerase chain reaction (PCR) were carried out to amplify the internal transcribed spacer (ITS) DNA using 100 ng of the extracted genomic DNA, 2 µM of magnesium chloride, Taq polymerase (1.25 units/50 µL of PCR), 1× Taq reaction buffer, 200 µM of dNTPs, and primer pairs (0.3 µM each) as listed in Table 10 below:

These primer pairs targeted the ITS regions located between 18S, 5.8S and 28S ribosomal RNA genes of the dinoflagellates. For each cycle of PCR, 1 min of denaturation was carried out at 94 °C, followed by 40s of primer annealing at 50 °C and then 2 min of extension at 72 °C. After thirty-five cycles, PCR was then ended by 5 min of 72 °C final extension and held at 4°C. PCR products were then purified by 1% agarose gel electrophoresis (140 V, 15 min). PCR products were poly(A)-tailed by adding Taq reaction buffer, Taq polymerase, magnesium chloride and dATP (1 mM) and incubated for 15 min at 72 °C. Finally, PCR products were sequenced by Sanger method with commercial facilities (BGI, Shenzhen, Guangdong, China). ITS DNA sequences were then identified by Basic Local Alignment Search Tool (BLAST).

### 4.3. Extraction and Quantification of Saxitoxins

To extract STXs in the dinoflagellates, 50 mL of dinoflagellate culture was harvested. Harvested cells were mixed with 300 µL of 0.5 M acetic acid prior to sonication (1 min, 24 kHz, 1 s/1 s pulse intervals). The samples were then transferred to Amicon^®^ Ultra 0.5 mL 10K Centrifugal Filters (MilliporeSigma, St. Louis, MO, USA) coupled with a collection tube. The samples were then centrifuged at 14,000× *g* for 20 min at 4 °C. Flowthrough was collected and subjected to column purification using Chromabond^®^ HILIC-SPE cartridges (Macherey-Nagel, Düren, Germany). The cartridges were preconditioned, equilibrated, and washed according to the standard protocol provided by the manufacturer. The samples were eluted with 3 mL of 0.2 M formic acid (FA) and dried by nitrogen gas. After that, samples were resuspended with 100 µL of 0.2 M FA. HILIC-purified samples and the standard of STXs (NRC, Ottawa, Ontario, Canada) were then separated and analyzed by Multiple Reaction Monitoring (MRM) in Sciex 6500 + LC/ESI—QTrap MS (AB Sciex LLC, Framingham, MA, USA) with reference to a previously established method [62].

### 4.4. Whole Lysate Extractions

For extracting whole lysate, 100 mL of cell cultures was harvested. Cell pellets were first cleaned by 1× phosphate-buffered saline (PBS; 137 mM NaCl, 2.7 mM KCl, 10 mM Na_2_HPO_4_, 1.8 mM KH_2_PO_4_, pH 8.0), then resuspended in 0.5 mL urea lysis buffer (8M urea, 2M thiourea, 65 mM dithiothreitol, 80 mM Tris) with 1% protease inhibitor cocktail P8340 (MilliporeSigma, St. Louis, MO, USA) containing bestatin hydrochloride, 4-(2-Aminoethyl)benzenesulfonyl fluoride hydrochloride (AEBSF), aprotinin, leupeptin hemisulfate salt, pepstatin A and N-(trans-Epoxysuccinyl)-L-leucine-4-guanidinobutylamide. Physical breakage of cells was carried out by using sonication for 1 min (24 kHz, 1s/1s pulse intervals, 4 °C), followed by spinning down of cell debris with refrigerated centrifugation at 12,000× *g* for 3 min at 4 °C. The supernatant was ready for protein quantification and sample preparation before examination by mass spectrometry.

### 4.5. Cell Surface Labeling

To distinguish the cell-surface proteins, an optimized cell-surface labeling protocol was developed to target the amino acids with primary amine that were exposed to the extracellular environment. About 200 mL of cell cultures was harvested and washed 3 times with ice-cold 2.5X PBS. The cell pellets were resuspended in 1.2 mL of 1× PBS to achieve a cell density of 2 × 106 cells/mL. One milliliter of 10mM sulfo-NHS-SS-biotin solution was freshly prepared by adding 6 mg of EZ-Link^TM^ sulfo-NHS-SS-biotin (ThermoFisher Scientific, Waltham, MA, USA) to 1 mL ultrapure water. One hundred and twenty-five microliters of sulfo-NHS-SS-biotin solution was added to each sample and incubated for 30 min at room temperature. First reaction of this reagent took place at the lysine residue or N-terminus of the cell surface protein because of the reactivity of the NHS moiety. Samples were centrifuged for 1 min at 12,000× *g* and at 4 °C and supernatants were removed. After the formation of intermediates and before cell rupture, the reaction was quenched immediately by adding a buffer containing primary amine to prevent unwanted labeling reaction on the inner cellular components. As a result, three washes of the cell pellets were performed by using 1 mL each of ice-cold 1× Tris buffered saline (TBS; 20 mM Tris, 150 mM NaCl, pH 8.0) with 1% protease inhibitors cocktail P8340 (MilliporeSigma, St. Louis, MO, USA). The labeling technique made use of mass add-on by disulfide bridge from the linker of the reagent that resulted in modification of the targeted residue with a net mass gain at 145.019 Da. With this chemical modification, the subcellular location of modified residues could be effectively revealed by mass spectrometry. Subsequently, the cell pellets were treated by the surface amphiesmal protein extraction protocol described in Section 4.6 below.

### 4.6. Surface Amphiesmal Protein Extractions

To enrich the amphiesmal proteins from the extraction, the labeled samples were firstly washed by 1× PBS and then resuspended in 500 µL of 1× PBS with 1% protease inhibitor cocktail P8340. Cells were ruptured by ultrasonication as described previously (Section 4.4). The supernatants were discarded, and cell pellets were resuspended and washed in 500 µL of 1× PBS with 1% protease inhibitor cocktail P8340 by centrifugation for 3 min at 12,000× *g* and 4 °C six times. After the washing procedure, the pellets were mixed with 200 µL of urea lysis buffer and 1% protease inhibitor cocktail, and then 1 min of sonication (24 kHz, 1s/1s pulse intervals, 4 °C) was performed. Subsequently, the samples were centrifuged for 10 min at 12,000× *g* and 4 °C and the resultant supernatants were collected for protein quantification and sample preparations for mass spectrometry.

### 4.7. Sample Preparation for Mass Spectrometry

One hundred microliters of protein extracts was reduced by 5 mM dithiothreitol (DTT) solution in 56 °C for 45 min, then alkylated by 15 mM iodoacetamide (IAA) at room temperature in the dark for 30 min. Afterwards, proteins were precipitated by 4 volumes of ice-cold acetone for 8 h at −20 °C. Precipitates were collected by centrifugation for 10 min at 12,000× *g*, 4 °C. Protein precipitates were air dried and then resuspended in 40 µL of urea buffer (8 M urea, 5 mM DTT, 150 mM Tris-HCl, pH 7.5). Twenty-five millimolar of ammonium bicarbonate was used to dilute the resuspended samples to a concentration of 1 µg/µL. Proteolysis of the sample was then carried out and the preparation and usage of proteases were used according to the protocol from the manufacturer. For all kinds of extracts, 4 µg trypsin V5280 (Promega, Madison, WI, USA) was added to each of the samples. For the surface labeled sample, an additional 5 µg of Glu-C V8 protease P6181 (MilliporeSigma, St. Louis, MO, USA) was added. Samples were incubated for 8 h at 37 °C. Subsequently, 5% trifluoroacetic acid (TFA) was added to the digested samples until a final pH at 2.0 was reached. The digested samples were then purified using Pierce™ C18 Spin Columns (ThermoFisher Scientific, Waltham, MA, USA) with the protocol suggested by the manufacturer. Samples were eluted in 80% acetonitrile (ACN) with 0.1% TFA, dried by refrigerated CentriVap Vacuum Concentrators (Labconco, Kansas City, MO, USA), and then re-dissolved in 0.1% FA.

### 4.8. RNA Extraction and Construction of Transcriptomic Databases for Proteomic Search

For total RNA extraction, cell pellets harvested from 100 mL of cultures of each species were homogenized by Precellys Evolution tissue homogenizer with Cryolys cooling (Bertin Technologies, Montigny-le-Bretonneux, France) in 4 °C with Trizol reagent (Hoffmann-La Roche, Basel, Switzerland). Trizol protocol was specially modified for dinoflagellate’s RNA extraction. Different from the original protocol, an extra step was added to remove excessive DNA contaminant from the samples. After the first three-phase separation, the resulting RNA-containing phase was transferred and mixed with another 1 mL of Trizol reagent for re-isolation. Then, the subsequent protocol remained the same as manufacturer’s recommendation. A total amount of 1 μg total RNA from each species was used for the cDNA library preparation.

cDNA libraries were prepared using NEBNext^®^ Ultra™ RNA Library Prep Kit for Illumina^®^ (New England BioLabs, Ipswich, MA, USA) following the polyA mRNA workflow recommended by the manufacturer. The clustering of the index-coded samples was carried out on cBot Cluster Generation System with PE Cluster Kit cBot-HS (Illumina, San Diego, CA, USA) and followed the instructions suggested by the manufacturer. After clustering, the library preparations were sequenced on the Illumina Hiseq 2000 platform and paired-end reads resulted as raw data. In addition, the paired-end reads were determined with a minimal read length at 200 bp. The fold coverage of the assemblies was at least 50X. After the removal of adaptor reads, poy-N containing reads and low-quality reads, clean data files from samples were further assembled, reconstructed, and clustered by Trinity [63]. Minimal count for K-mers to be assembled (i.e., min_kmer_cov) was set at 2 by default and default in all other parameters. A transcriptomic database for each species was built for proteomic search.

### 4.9. Nano-LC Coupled ESI-Orbitrap-MS/MS and Proteomic Search

Samples were separated and analyzed by Orbitrap Fusion Lumos Mass Spectrometer coupled with UltiMate™ 3000 RSLCnano system and nanoelectrospray ionization system (ThermoFisher Scientific, Waltham, MA, USA). Specifically, 0.1 µL of samples was first injected into a 15-cm C18 nano-flow column (LC Packings, Waddinxveen, The Netherlands). Peptides were eluted by a 6–30% gradient of 0.1% FA in ACN. The flow rate was kept at 0.3 µL/min and 120 min gradient program was used.

Eluted peptides were introduced directly to the nano-ESI-Orbitrap MS operating in positive mode with 2100 V capillary voltage and 300 °C of ion transfer tube temperature. Samples were analyzed in data-dependent acquisition (DDA) mode of tandem mass spectrometry in the Orbitrap analyzer. The main settings are described in Table 11 below:

Raw data acquired by the Orbitrap-MS were then analyzed using Mascot Distiller 2.5 (Matrix Science, London, UK) and searched in our in-house constructed transcriptomic databases via Mascot Server 2.4 (Matrix Science, London, UK). Parameters for the decoy search were 1% False Discovery Rate (FDR) at identity level, peptide mass tolerance at ±15 ppm, ion score cut-off at 0.01 and one missed cleavage was allowed. The protein modifications were set up as listed in Table 12 below:

On the other hand, spectra were also input into Progenesis QIP (Waters, Milford, MA, USA) for Principal Component Analysis (PCA) and automatic processing mode was used for peak picking and peptide ion detection. Principal components between mass spectra were then predicted and compared.

### 4.10. Accession Numbers

The 15 raw data files of the cDNA libraries were deposited in the National Center for Biotechnology Information (NCBI) as NCBI Sequence Read Archive (SRA) under the BioProject PRJNA634431: Dinophyceae sp. Raw sequence read (TaxID: 1918384).

The link for reviewer is shown as follows: https://dataview.ncbi.nlm.nih.gov/object/PRJNA634431?reviewer=vse9qbbk1adoht1kjgtmsj148v (accessed on 3 September 2021).

## Figures and Tables

**Figure 1 toxins-13-00624-f001:**
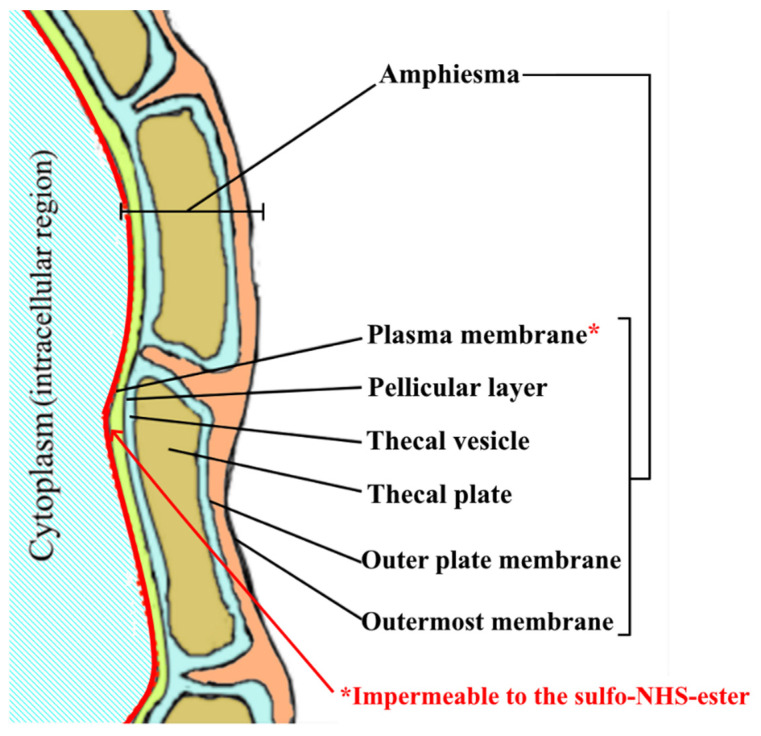
Schematic diagram of the amphiesmal organization in thecate dinoflagellate. Note—This diagram is modified from Morrill and Loeblich [24], and Kwok and Wong [14].

**Figure 2 toxins-13-00624-f002:**
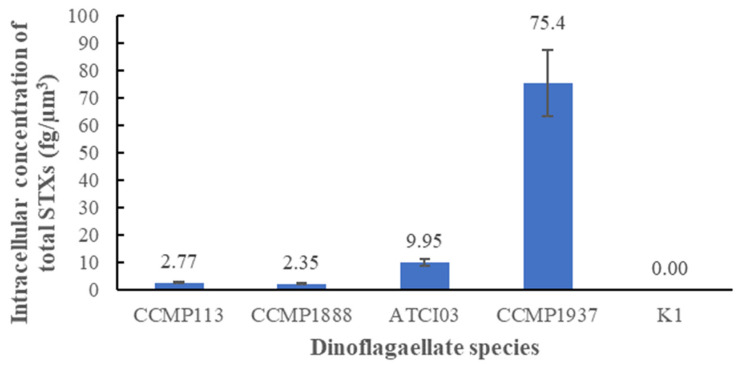
Intracellular concentration of total STXs in the five species.

**Table 1 toxins-13-00624-t001:** BLAST results of the ITS regions of the selected dinoflagellate species.

		BLAST Search Results			
Species	Primer *	Accession No. of Best Match	% Ident.	E-Value	Source Organism
CCMP113	Pair1	FJ823523.1	99	0	*Alexandrium minutum*
	Pair2	JF521634.1	99	0	*Alexandrium minutum*
CCMP1888	Pair1	JF906999.1	99	0	*Alexandrium lusitanicum*
ATCI03	Pair2	JF906992.1	99	0	*Alexandrium tamarense*
CCMP1937	Pair1	DQ779989.2	99	0	*Gymnodinium catenatum*
K1	Pair1	KU314866.1	99	0	*Karenia mikimotoi*
	Pair2	LC055227.1	99	0	*Karenia mikimotoi*

* For details of primers, please refer to Section 4.2. Note—The DNA sequences of the PCR products are listed in Appendix A.

**Table 2 toxins-13-00624-t002:** Overall statistics of transcriptome-guided orthogroup inference.

Statistical Categories	Quantity
**Genes inputted**	
No. of genes	695,887
No. of genes in orthogroups	386,478
No. of unassigned genes	309,409
No. of genes in species-specific orthogroups	6749
**Orthogroup inferred**	
No. of orthogroups	86,675
No. of species-specific orthogroups	1539
Specific to CCMP113	35
Specific to CCMP1888	36
Specific to ATCI03	201
Specific to CCMP1937	581
Specific to K1	686
No. of orthogroups commonly found in STXs-producing species	3204
No. of orthogroups commonly found in *Alexandrium* species	11,136
No. of orthogroups with all species present	14,338
No. of single-copy orthogroups	3234
Mean orthogroup size	4.5
Median orthogroup size	3
G50 (assigned genes)	5
G50 (all genes)	2
O50 (assigned genes)	20,072
O50 (all genes)	67,408

**Table 3 toxins-13-00624-t003:** Percentage overlap of orthogroups among the species.

	CCMP113	CCMP1888	ATCI03	CCMP1937	K1
**CCMP113**	100	-	-	-	-
**CCMP1888**	95	100	-	-	-
**ATCI03**	80	80	100	-	-
**CCMP1937**	65	64	53	100	-
**K1**	65	65	73	74	100

**Table 4 toxins-13-00624-t004:** Statistics of proteins identified in surface amphiesmal protein extractions (SAPEs) by LC-MS and Mascot search.

	CCMP113	CCMP1888	ATCI03	CCMP1937	K1
**No. of protein identified in SAPEs**					
**Amphiesmal protein ^1^**	180	217	203	218	197
▪ Labeled	44	43	23	21	36
○ Transporter protein	4	5	2	1	4
▪ Unlabeled ^2^	136	174	180	197	161
**Non-amphiesmal protein ^3^**	286	339	309	276	314
▪ Labeled	51	25	30	14	79
▪ Unlabeled	235	314	279	262	235
Unknown protein	154	211	131	210	160
**Total**	620	767	643	704	671
**Percentage yield (%)**					
**Amphiesmal protein ^1^**	29.0	28.3	31.6	31.0	29.4
▪ % Yield compared to WL ^4^	+6.56	+2.33	+6.71	+5.50	+4.23
▪ % Yield compared to ICF ^4^	+1.36	+1.33	+2.95	+2.23	0
**Labeled amphiesmal protein ^1^**	7.10	5.61	3.58	2.98	5.37

^1^ Amphiesmal proteins were defined as proteins annotated with any of the following amphiesmal GO terms: apoplast (GO:0048046), cell surface (GO:0009986), cell wall (GO:0005618) and membrane (GO:0016020). ^2^ Unlabeled amphiesmal proteins referred to the proteins that were identified with amphiesmal GO terms, but the isotopic tag (+145.019 Da) was not found in their corresponding peptide peaks in the MS spectra. ^3^ Non-amphiesmal proteins were defined as proteins identified without amphiesmal GO terms but other subcellular locations such as cytoplasm (GO:0005737) and cytosol (GO:0005829). ^4^ For the details of proteins identified in WL and ICF, please refer to Appendix A.

**Table 5 toxins-13-00624-t005:** Transporter homologs labeled and identified in the amphiesma of dinoflagellates.

Species	Uniprot BLASTp Results #		GO-Terms		Unigene IDs of Protein Identified *	No. of Peptide Species Identified in MS
	E-Value (Respective Accession No. of Best Match)	Annotation of Homolog	Cellular Component	Biological Process		
**CCMP113**	0 (P499510) 0 (Q2RBN7)	Clathrin heavy chain 1	Endosome (GO:0005768), Membrane (GO:0016020)	Transferrin transport (GO:0033572), Receptor internalization (GO:0031623), Intracellular protein transport (GO:0006886)	Cluster-16584.55282 Cluster-16584.53449	5 4
	4.1 × 10^−198^ (Q61941)	NAD(P) transhydrogenase, mitochondrial	Membrane (GO:0016020)	Proton transmembrane transport (GO:1902600)	Cluster-16584.47837	6
	7.7 × 10^−^^202^ (P80021)	ATP synthase subunit alpha, mitochondrial	Plasma membrane (GO:0005886)	ATP synthesis coupled proton transport (GO:0015986)	Cluster-16584.49485	14
**CCMP1888**	4.1 × 10^−198^ (Q61941) 1.5 × 10^-183^ (Q61941) 1.6 × 10^−148^ (P11024) 1.0 × 10^−^^138^ (P11024)	NAD(P) transhydrogenase, mitochondrial	Membrane (GO:0016020)	Proton transmembrane transport (GO:1902600)	Cluster-6994.52752 Cluster-6994.52754 Cluster-6994.53793 Cluster-6994.53135	7 5 3 2
	7.7 × 10^−202^ (P80021)	ATP synthase subunit alpha, mitochondrial	Plasma membrane (GO:0005886)	ATP synthesis coupled proton transport (GO:0015986)	Cluster-6994.53837	20
**ATCI03**	1.8 × 10^−138^ (Q8DHU4)	Protein translocase subunit SecA	Plasma membrane (GO:0005886)	Protein import (GO:0017038)	Cluster-15238.31582	7
	8.4 × 10^−190^ (P11024)	NAD(P) transhydrogenase, mitochondrial	Membrane (GO:0016020)	Proton transmembrane transport (GO:1902600)	Cluster-15238.36601	3
**CCMP1937**	0 (Q00610)	Clathrin heavy chain 1	Endolysosome membrane (GO:0036020), Plasma membrane (GO:0005886)	Membrane organization (GO:0061024), Intracellular protein transport (GO:0006886)	Cluster-5567.56526	7
**K1**	1.3 × 10^−97^ (Q13423) 1.5 × 10^−97^ (Q13423)	NAD(P) transhydrogenase, mitochondrial	Membrane (GO:0016020)	Proton transmembrane transport (GO:1902600), Tricarboxylic acid cycle (GO:0006099)	Cluster-24592.79099 Cluster-24592.45845	8 7
	7.8 × 10^−198^ (Q03265) 3.0 × 10^−163^ (Q03265)	ATP synthase subunit alpha, mitochondrial	Cell surface (GO:0009986)	ATP synthesis coupled proton transport (GO:0015986)	Cluster-24592.43084 Cluster-24592.74857	13 8

# Estimated by the best score and E-value of the matches in BLASTp results. * The amino acid sequences, individual mascot score of each protein identified and the position of labeled residue are shown in Appendix A. Note—Mascot score of all these protein matches reached the confidence level > 95% (i.e., Mascot score > 90).

**Table 6 toxins-13-00624-t006:** Common homologs of transporter identified in multiple species.

Name of Transporter Homologs	Dinoflagellate Species				
	CCMP113	CCMP1888	ATCI03	CCMP1937	K1
NAD(P) transhydrogenase, mitochondrial	✓	✓	✓	--	✓
ATP synthase subunit alpha, mitochondrial	✓	✓	--	--	✓
Clathrin heavy chain 1	✓	--	--	✓	--

Note—“✓” represented the presence of the transporter homologs while “--” represented the absence of the transporter homologs in the SAPEs of the species

**Table 7 toxins-13-00624-t007:** Results of orthogroup assignment for the proteins identified.

	Dinoflagellate Species				
	CCMP113	CCMP1888	ATCI03	CCMP1937	K1
**No. of protein identified in SAPE**					
**Assigned to orthogroups**	532	648	383	541	384
▪ Species-specific orthogroups	1	0	0	1	11
▪ STXs-producing species-specific orthogroup	7	7	7	7	N/A
**Unassigned to orthogroups**	88	119	119	163	287
**Total**	**620**	**767**	**502**	**704**	**671**

**Table 8 toxins-13-00624-t008:** Species-specific proteins identified in SAPEs.

Species	Uniprot BLASTp Results #		GO-Terms		Unigene IDs of Protein Identified *	No. of Peptide Species Identified in MS
	E-Value (Respective Accession No. of Best Match)	Uniprot Annotation of Ortholog	Cellular Component	Biological Process		
**CCMP113**	7.9 × 10^−277^ (P16100)	Isocitrate dehydrogenase [NADP]	Cytoplasm (GO:0005737)	Tricarboxylic acid cycle (GO:0006099)	Cluster-16584.46230	1 ^
**CCMP1937**	8.7 × 10^−92^ (P91622)	Pyruvate dehydrogenase kinase	Mitochondrion (GO:0005739)	Glucose metabolic process (GO:0006006)	Cluster-5567.50671	3
**K1**	1.2 × 10^−7^ (Q39709)	Fucoxanthin-chlorophyll a-c binding protein	Chloroplast thylakoid membrane (GO:0009535)	photosynthesis, light harvesting (GO:0009765)	Cluster-24592.102021	4
	3.1 × 10^−36^ (Q40296) 3.1 × 10^−38^ (Q40297) 1.3 × 10^−34^ (Q40296) 7.8 × 10^−22^ (Q40296)	Fucoxanthin-chlorophyll a-c binding protein B	Chloroplast thylakoid membrane (GO:0009535)	photosynthesis, light harvesting (GO:0009765)	Cluster-24592.122404 Cluster-24592.47428 Cluster-24592.64372 Cluster-24592.96057	5 2 2 6
	2.0 × 10^−27^ (Q40300)	Fucoxanthin-chlorophyll a-c binding protein F	Chloroplast thylakoid membrane (GO:0009535)	photosynthesis, light harvesting (GO:0009765)	Cluster-24592.21292	2
	2.2 × 10^−^^20^ (A0A5C6NSI7)	Dual specificity mitogen-activated protein kinase 4	protein phosphorylation (GO:0006468)	--	Cluster-24592.69574 Cluster-24592.107035	1 ^ 1 ^
	5.0 × 10^−63^ (M2X807)	CAAX amino terminal protease family protein	Integral component of membrane (GO:0016021)	CAAX-box protein processing (GO:0071586)	Cluster-24592.117554	2

# Estimated by the best score and E-value of the matches in BLASTp results. ^ These peptides contained at least 15 amino acids and were identified with peptide score > 50. Please refer to Appendix A for the peptide score of these peptides. * The amino acid sequences, individual mascot score of each protein identified and the position of labeled residue are shown in Appendix A. Note—Mascot score of all these protein matches reached the confidence level > 95% (i.e., Mascot score > 90).

**Table 9 toxins-13-00624-t009:** STX-producing-species-specific orthologs identified in SAPEs.

Orthogroup No.	Uniprot Annotation of Ortholog # (Accession No.)	GO-Terms		No. of Peptide Species Identified in MS			
		Cellular Component	Biological Process	CCMP113	CCMP1888	ATCI03	CCMP1937
OG0004814	Mitochondrial dicarboxylate/tricarboxylate transporter (Q9C5M0, P0C582)	Mitochondrion inner membrane (GO:0005743)	Oxoglutarate:malate antiporter activity (GO:0015367)	1 ^	2	7	2
OG0024578	Glutamine synthetase (P15623, Q54WR9)	phagocytic vesicle (GO:0045335)	Nitrogen compound metabolic process (GO:0006807)	1 ^	1 ^	1 ^	2
OG0000905	Fucoxanthin-chlorophyll a-c binding protein E (Q40301)	Chloroplast thylakoid membrane (GO:0009535)	Light-harvesting complex (GO:0030076)	16	10	9	16
OG0034291	F-type H^+^-transporting ATPase subunit beta (A6BM09, Q06J29)	Chloroplast thylakoid membrane (GO:0009535)	Proton-transporting ATP synthase activity (GO:0046933)	22	28	32	25
OG0034855 *	F-type H^+^-transporting ATPase subunit alpha (Q9MUT2, Q1ACM8, Q85X67)	Chloroplast thylakoid membrane (GO:0009535)	Proton-transporting ATP synthase activity (GO:0046933)	13	17	22	19
OG0035644	Pentatricopeptide repeat-containing protein (Q9SV46, Q9SZ52, Q9M907, Q9LVQ5)	Chloroplast (GO:0009507)	RNA stabilization (GO:0043489) Translation (GO:0006412)	10	7	11	14

# Estimated by the best score and E-value of the matches in BLASTp results. ^ These peptides contained at least 15 amino acids and were identified with peptide score > 50. Please refer to Appendix A for the peptide score of these peptides. * Successfully labeled in all four STX-positive species. The amino acid sequences, individual mascot score of each protein identified and the position of labeled residue are shown in Appendix A. Note—Mascot score of all these protein matches reached the confidence level > 95% (i.e., Mascot score > 90).

**Table 10 toxins-13-00624-t010:** Primer sequences used for ITS DNA-targeted PCR.

Primer	Forward (5′-3′)	Reverse (5′-3′)
Pair 1	TCCGTAGGTGAACCTGCGG	TCCTCCGCTTATTGATATGC
Pair 2	TGAACCTTAYCACTTAGAGGAAGGA	GCTRAGCWDHTCCYTSTTCATTC

**Table 11 toxins-13-00624-t011:** Program setting of the ionizer and Orbitrap analyzer.

Parameters	DDA-MS1	DDA-MS2
Resolution	60,000	15,000
Scan range (m/z)	350–1500	Auto
Max. injection time (ms)	20	30
AGC target	4.0 × 10^5^	5.0 × 10^4^
HCD collision energy (%)	N/A	30
Intensity threshold	1.0 × 10^4^	N/A

**Table 12 toxins-13-00624-t012:** List of Variable modifications for the proteomic search in Mascot databases.

Variable Modification	ΔM.W. (Da)	Targeted Amino Acid
Carbamidomethyl	+57.021	C
Oxidation	+15.994	M
3-[(Carbamoylmethyl)sulfanyl]propanoyl #	+145.019	K, Protein N-term

# This modification was specifically applied to process the mass spectra of SAPEs.

## Data Availability

The RNA-seq datasets analyzed in this study is scheduled to be publicly available after 30 June 2022. These datasets (NCBI BioProject: PRJNA634431) can be found here: https://dataview.ncbi.nlm.nih.gov/object/PRJNA634431?reviewer=vse9qbbk1adoht1kjgtmsj148v (accessed on 11 July 2021).

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
