# Peer review of "Finding Species-Specific Extracellular Surface-Facing Proteomes in Toxic Dinoflagellates"

_toxins, 2021, doi:10.3390/toxins13090624_

Round 1

Reviewer 1 Report

This study developed a method to label and capture proteins on the cell surface of dinoflagellates. They used five different species of dinoflagellates who produce toxins and also produced transcriptomes for each of these species.

Overall, I think this is a very neat and useful method. The following are my different comments from reading the manuscript.

I firstly think the English needs to be read over/improved with a native English speaker. In particular I think the verb tenses need to be evaluated throughout the article.

The authors claim that Gymnodinium catenatum is a thecated species. I am wondering if there is a paper that claims that it does in fact have theca? To my knowledge, G. catenatum is naked/unarmored and thus will impact your results significantly.

Secondly, Karenia mikimotoi contains a different plastid from the other four being studied here, which may help with page 17 findings in table 7.

How were the transcriptomes created? No detail in the methods.

Were the non-amphiesmal proteins in table 4 labeled using your method?

Why do you think STX transporter was not labeled/found? The method is not sensitive enough or?

At the end I guess I am left wondering what the method can be used with going forward. The intro now made me think it can be used for toxic species identification but now I do not think that is the case. I think reworking the narrative could help the story/usefulness of the method.

Reviewer 2 Report

This manuscript describe the novel methodology for labeling surface protein and identification of them using LC-MS/MS strategy. The combination of these methods is  unique. However, the identified proteins may find ubiquitously in dinoflagellates and may not be used as marker for identification of the species mentioned in the manuscript. 

The data were obtained and analyzed properly and the design of the experiments seem to be fine. Thus, this manuscript should be accepted to the journal since the manuscript contains many interesting features which should attract the readers of the journal. 

I have one question that is there any "non-specific reaction" of the labeling reagent? Authors commented that sulfo-NHS-SS-biotin would not pass through plasma membrane in the main text. However, the reagent should pass through the membrane in some extent and the reagent might react with non-protein molecule in the outer surface of the dinoflagellates.  Authors advised to examine the non-specific reaction and show the data in the manuscript.  

Round 2

Reviewer 1 Report

Thank you for the additional material I think the manuscript has greatly improved.

The only main issue I have is when you are comparing the STX producing species with the non-STX species when it comes to anything regarding the plastid. Since Karenia is known to have a different plastid/rubisco it is important you do not claim that the difference you are seeing is from STX producing vs non-STX (see https://doi.org/10.3732/ajb.91.10.1523). If you want to do these comparisons it would be ideal to acquire non-STX Alexandrium species or Gymnodinium species.
